# Similarity in Temporal Movement Patterns in Laying Hens Increases with Time and Social Association

**DOI:** 10.3390/ani12050555

**Published:** 2022-02-23

**Authors:** Yamenah Gómez, John Berezowski, Yandy Abreu Jorge, Sabine G. Gebhardt-Henrich, Sabine Vögeli, Ariane Stratmann, Michael Jeffrey Toscano, Bernhard Voelkl

**Affiliations:** 1Center for Proper Housing: Poultry and Rabbits (ZTHZ), Division of Animal Welfare, VPH Institute, University of Bern, Burgerweg 22, 3052 Zollikofen, Switzerland; sabine.gebhardt@vetsuisse.unibe.ch (S.G.G.-H.); sabine.voegeli@blw.admin.ch (S.V.); ariane.stratmann@vetsuisse.unibe.ch (A.S.); michael.toscano@vetsuisse.unibe.ch (M.J.T.); 2Veterinary Public Health Institute, University of Bern, 3012 Bern, Switzerland; john.berezowski@vetsuisse.unibe.ch; 3National Centre for Animal and Plant Health, San José de las Lajas 32700, Cuba; yandiru@nauta.cu; 4Division of Animal Welfare, VPH Institute, University of Bern, 3012 Bern, Switzerland; bernhard.voelkl@vetsuisse.unibe.ch

**Keywords:** social network, phenotype variation, individuality, movement pattern, behavioural plasticity

## Abstract

**Simple Summary:**

Social bonds are well-known to affect individual decisions or responses in specific situations, however, the impact of sociality on the individual activity behaviour of hens is largely unexplored. Therefore, we explored the relationship between social associations and individual daily movement patterns by studying the range use of domestic hens in a semi-commercial barn setting with automatic tracking of laying hens. We analysed the social relationships and the daily movement patterns of hens and found that hens were consistent in their individual variation in daily movement patterns and maintained stable social relationships. The social associations among hens correlated with movement patterns and this correlation increased with time, leading to more similar movement patterns of socially associated individuals. This study clearly shows that the social environment within a group can shape and enhance variation in movement patterns of individual animals.

**Abstract:**

We explored the relationship between social associations and individual activity patterns in domestic hens. Out of 1420 laying hens, 421 hens were equipped with RFID tags attached to RFID-specific leg bands (leg bands from Company Roxan, Selkirk, Scotland) to continuously track their change in location across four different areas (one indoor and three outdoor areas). Using a combination of social network analysis for quantifying social relationships and dynamic time warping for characterizing the movement patterns of hens, we found that hens were consistent in their individual variation in temporal activity and maintained stable social relationships in terms of preferred association partners. In addition to being consistent, social associations correlated with movement patterns and this correlation strengthened over the period of observation, suggesting that the animals aligned their activity patterns with those of their social affiliates. These results demonstrate the importance of social relationships when considering the expression of individual behaviour. Notably, differences in temporal patterns emerge despite rather homogeneous rearing conditions, same environment, and low genetic diversity. Thus, while variation in behavioural phenotypes can be observed across isolated individuals, this study shows that the social environment within a group can shape and enhance variation in general movement patterns of individual animals.

## 1. Introduction

Social organization is a key aspect of animal ecology, closely interlinked with anatomical, morphological, physiological, and behavioural traits [1,2,3]. Recently, researchers have highlighted the intrinsic interrelationship between social relationships among individuals and their small- and large-scale activity patterns [4,5,6]. Resource distribution can affect the social composition of animal groups via individuals’ behaviour [7,8,9,10], while, on the other hand, maintaining social relationships can lead to individuals refraining from maximising resource acquisition [11]. Specifically, if groups of animals are moving, group cohesion represents a permanent challenge as individuals have to coordinate their movements and more generally their activity patterns. How this feat can be achieved has been studied under the umbrella term of collective behaviour [12,13].

The presence of many individuals in a small area can lead to various forms of competition—e.g., when individuals interfere with each other’s foraging activity [14]—or sometimes kleptoparasitism [15,16,17]. In such cases, composition, size and structure of the social group have direct consequences on the effectiveness of foraging activity. Foraging is not the only activity affected by social cohesion of pair- or group-living animals. Time budgets for many types of activities are substantially affected as individuals have to accommodate time for social interactions [18]. For example, under aversive conditions (e.g., food scarcity) which require increased foraging activity, baboons (*Papio* sp.) reduce resting time rather than social grooming time—the latter being an important mechanism for forming and maintaining social bonds [1,19].

While it has been shown that social bonds can affect individual decisions or responses in specific situations e.g., [20,21], the impact of sociality on the daily activity period is largely unexplored. The lack of studies is partly due to the difficulties in tracking individual animals within large groups throughout the entire day and also due to difficulties in comparing complex movement patterns using quantitative analyses. In order to explore how social relationships interact with the temporal structuring of individual behaviour we studied the ranging patterns of domestic hens (*Gallus domesticus*) in a semi-commercial barn setting that provides several outside ranging areas for the animals to access. We wanted to know whether individual differences in daily activity patterns, as they have been described for domestic hens [22,23,24,25], are reflected in social associations between birds. More specifically, are more closely associated individuals within a group also more similar in their daily movement patterns and do these patterns become assimilated over time in accordance with social bonds? To examine this, we (i) established whether hens formed non-random social associations that can be interpreted as individualized social relationships, (ii) used time series analysis in order to investigate whether hens show consistent individual differences in their daily activity patterns, and (iii) asked whether similarities in daily activity patterns were correlated with social associations. As this study is, to the best of our knowledge, the first study investigating such a relationship between social association and activity patterns, it is primarily of an exploratory nature.

## 2. Materials and Methods

### 2.1. Ethical Approval

This study was approved by the Cantonal Office of Berne (BE-46/16) and met all cantonal and federal regulations on animal experimentation.

### 2.2. Animals and Housing

Commercial Brown Nick laying hens (*Gallus gallus domesticus*) were maintained at the Aviforum (Zollikofen, Switzerland), which is a poultry research facility, intended to provide commercially relevant conditions for experimental trials. All birds were reared together from hatch until 17 weeks of age in eight pens (600 hens/pen). At 17 weeks of age, 1420 birds were transferred to four pens of a commercial laying hen house (355 birds/pen). The four pens (each 12.9 m^2^) contained a Rihs Bolegg II commercial aviary system (Krieger AG, Ruswil, Switzerland) with a maximum total stocking density of 9.33 hens/m^2^ and group nests provided along one side of the walls. Birds transferred from each rearing pen were proportionally distributed across all eight laying pens consisting each of a stacked-tier aviary. Detailed description of the aviary structure can be found elsewhere as part of a related effort with the same study animals [26].

The pen floors were covered with 10 cm of wood shavings. Birds had access to three different outdoor areas outside of the barn (Figure 1): a covered outdoor area (“wintergarden” (WG), average size: 17.55 m^2^) equipped with litter, nipple drinkers and perches; a fenced area containing small stones (“stone yard” (SY), average size per pen: 88 m^2^); and a pasture (“free-range area” (FR), average size per pen: 288 m^2^). Each pen and its associated outdoor area was divided by fencing to maintain separate populations. Within a pen, each area was also divided by fencing (or the barn wall) to restrict access to some but not all outdoor areas. Birds could only move between areas (i.e., inside, wintergarden, stone yard, free-range) via a single location between them (pop hole or gate: minimum dimensions: 1 m width × 0.22 m depth × 0.45 m height) that provided unobstructed access when opened. Transitions between areas could only occur between two juxtaposed areas (Figure 1), e.g., transitioning directly from the barn to the FR without passing through the WG and SY was not possible. Artificial light was provided by LED (Light Emitting Diode) lighting in the barn from 02:00 to 17:00 h with a transitional phase of 5 min beginning at 02:00 h and 15 min at 16:45 h. Natural daylight was provided within the barn from 08:00 to 16:30 h through windows controlled by curtains. Birds were encouraged to return into the barn interior (shooed in) at around 16:30 h.

### 2.3. Data Acquisition

The experimental study was conducted after the birds were transferred to a semi-commercial layer barn, where they had access to different types of outdoor areas (wintergarden, stone yard, free-range). Prior to the transfer of the birds, two sets of antennae (Gantner Pigeon System, http://www.benzing.cc/, accessed 13 February 2022) were positioned on both sides of the transition locations (pop-holes and gates) between two areas (e.g., inside/wintergarden) to ensure that hens passed over both sets of antennae when moving between areas. The installation in this manner led to two registrations per bird for each transition, i.e., entering into one area and exiting another. The installation took place before the hens were transferred at 17 weeks of age, therefore no habituation to the RFID antennae and protection was needed. A more detailed description of the RFID system, installation and operation are described by Guerrero-Bosagna et al. [26]. The system reliability was described by Gebhardt-Henrich et al. [22]. A detailed documentation of all steps of data curation is given in the Appendix A.

A total of 110 hens per pen were selected in a stratified manner during the transfer from the rearing pens to the laying pens by picking every third bird per transport crate (depending on the total number of birds per crate, this was: 3 × 7 animals per box, 14 × 6 animals per box, 1 × 5 animals per box = 110 birds/pen) and fitting them with radio frequency identification (RFID) tags (Hitag S 2048 bits, 125 kHz, diameter: 4.0 mm, length: 34.0 mm). The RFID tags were attached to adjustable RFID-specific leg bands (Company Roxan, Selkirk, Scotland), as described by Gebhardt-Henrich et al. [22]). Due to various bacterial infections, birds were prevented from leaving the inside area for the first 60 days after application of the RFID tags and those days were not considered in the analyses. Furthermore, due to some drop out reasons (e.g., death of focal birds, lost RFID tags, broken devices etc.) the analysis of the movement patterns was restricted to the 421 hens with complete continuously collected tracking data.

Movement patterns were recorded between the 10th of May and 23rd of October 2016, a time span of 166 days. However, access to all areas was only provided on 72 days throughout this time span. Because we were only interested in the usage of different outdoor areas (recording the transition across gates/pop holes), investigations and analyses were restricted to those 72 days only. Recordings started in the morning, when the pop-holes of the barn that gave access to the three different outdoor areas (wintergarden, stone yard and free-range; Figure 1) were opened and ended in the afternoon, when the pop-holes were closed. If animals were still in the outside areas at the time of closing the pop-holes, they were shooed in by the animal care takers. The action of forcing animals inside affected the behaviour of the animals, though as only 0.16 % of the transitions occurred during the final five minutes of the day, we considered the potential effect on the overall results negligible.

In order to link movement patterns to regional weather conditions, such as temperature, data from the nearest regional weather station (Belp Airport, Bern, Switzerland (LSZB), 10.2 km away) was used. Over the observation period, the maximum daily temperature was between 7 and 32 degrees Celsius and rain was recorded on 12 days. Details on the influence of weather conditions on bird activity are given in the online material (Appendix A).

### 2.4. Movement Data

Based on recordings at the antennas, for all individuals we constructed lists of time-stamped transitions between areas, lists of sojourn times in the areas (duration per stay in the areas), and continuous time series indicating the individuals’ positions based on transition events. Missing data due to the birds RFID tag not being detected by an antenna were dealt with differently for the respective lists. For sojourn times, events were only recorded where both the entry as well as the exit were recorded. For the time series analysis it was necessary to construct a continuous data stream; thus whenever an entry was missing and the exact time of transition from one area to the next could not be determined, the transition time was interpolated to be half-way between the entry of the first area and the exit from the second area. In order to investigate whether transitions between areas happened in a coordinated manner, with times where groups of hens would move from one area to another and times with no transitions, we compared the observed times between two transitions of hens from one area to an adjacent area with expected values assuming an even distribution of transition events. Expected values for each day and each antenna were estimated by randomly sampling *k* transition times from a uniform distribution U(*a*,*b*), where *k* was the number of observed transitions, *a* the time stamp of the first transition and *b* the time stamp of the last transition. Furthermore, we calculated a leading index for each individual by building the proportion of all transitions of the respective hen, where the time gap between the preceding hen and the focus hen was larger than the time gap between the focus hen and the next hen to pass through the same gate.

Finally, for each individual we calculated a descriptive measure for overall activity: the proportion of time spent in the four different areas, the number of days on which the animals were registered in one of the areas outside the barn, the order in which animals left the wintergarden in the morning (WG to SY), the order in which the animals returned from the stone yard in the afternoon (SY to WG), the total number of transitions between all areas regardless of direction, the sample entropy as a measure of predictability [27] in the daily movement patterns, and a leading index, which is a metric that indicates whether a bird was more frequently following other birds or whether it was more often followed by others (Appendix A).

### 2.5. Social Network

In order to create social networks, we identified dyadic associations defined as spatio-temporal co-occurrences of birds at the RFID antennas fixed at the transition point between neighbouring areas (i.e., between IN and WG, WG and SY, and SY and FR). If a bird was registered at one of these locations within 5 s of one or more other birds, we considered this to be a dyadic spatio-temporal co-occurrence and added a corresponding edge to the co-occurrence matrix. Associations were, thus, recorded as point events. Adding an edge each time we observed a co-occurrence at an antenna resulted in a symmetric multi-edge graph, with vertices representing individuals, edges associations (based on co-occurrence) between individuals and the number of edges indicating how often two hens co-occurred at an antenna. The total number of movements between areas, and hence registration at the antennas, differed between hens. As this also influenced the likelihood of co-occurrences by chance, we calculated an association index (AI)
AI(AB) = AB/(A + B − AB), 
where A is the number of registrations of bird A, B is the number of registrations of bird B, and AB is the number of co-occurrences of birds A and B [28]. The resulting association matrix was a symmetric weighted matrix. We calculated association matrices separately for each day as well as for the entire study period.

### 2.6. Dynamic Time Warping Analysis

Dynamic time warping (DTW) is a shape-based method for comparing the dissimilarity (a dimensionless quantity) of two time series, independent of their individual length. The DTW approach matches similar shapes between two time series that vary in their length, even when shifted in the time dimension. Calculation of a DTW value results in lower dissimilarity distances for time series which contain similar shapes that are shifted in time, compared with methods that estimate Euclidian distances between individual time points in a time series [29]. The lower the DTW dissimilarity distance value the higher the similarity between two time series. The analyses with the dissimilarity distance matrices were all conducted in R [30]. The DTW function used in this study (“dtwclust”, [31]) creates a dissimilarity distance matrix with one dissimilarity value for each pairwise time series comparison. All possible pairs of time series were compared, including comparisons of all daily time series for individual hens and all comparison of all daily time series between hens. This allows the resulting dissimilarity distance matrix to be used to cluster hens with similar spatio-temporal ranging patterns irrespective of the exact transition times.

Dynamic time warping was conducted separately for each of the four pens. The dissimilarity distance matrix resulting from DTW was a 4-dimensional *a* × *b* × *a* × *b* hypermatrix (M_4_), where *a* is the number of hens in the pen and *b* is the number of tracking days. Each element *dd_i,j,k,l_* of M_4_ is the dissimilarity distance between the time series (*ts_i,j_*) of hen *i* on day *j* and the time series (*ts_k,l_*) of hen *k* on day *l*, *dd*(*ts_i,j_*, *ts_k,l_*).

To investigate the consistency in movement patterns, the dissimilarity distances were considered per hen to allow within-hen and between-hen comparisons. For the within-hen comparison, we calculated for each hen *i* the median for the dissimilarity distances (*dd_i,j,k,l_*):{*dd_i,j,k,l_* ϵ M_4_|*i* = *k &* 1 ≤ *j* ≤ *b* & 1 ≤ *l* ≤ *b* & *l* > *j*}.

For the between-hen comparison for each hen *i* to all other hens in the pen the median was calculated for the dissimilarity distances:{*dd_i,j,k,l_* ϵ M_4_|*i* ≠ *k &* 1 ≤ *j* ≤ *b* & 1 ≤ *k* ≤ *a* & 1 ≤ *l* ≤ *b*}.

For the statistical analyses a Wilcoxon signed rank test was conducted for each of the four pens.

To estimate the long-term trend in daily dissimilarity distances of the 72 tracking days within the entire tracking period of 166 days on which all outdoor areas were accessible, generalized additive models with days as a smoothing factor were created using the R package “mgcv” (version: 1.8–36) [32]. Because of the high consistency of pairwise dissimilarity distances found across time according to the day-to-day differences per hen and to the daily mean of all pairwise DTW distances of all hens, a 2-dimensional a x a matrix A was extracted. This was done by summing for each pair of hens all dissimilarity distances on a given day over all days, where the element A*_i_*_,*k*_ of the matrix A equals:$$\sum_{j=l=1}^{b}M_{i,j,k,l}.$$

This summed matrix was then used for the hierarchical clustering to visually group hens. For this approach the Diana method (Divisive Analysis using the R package “dtwclust”, [33]) was employed using the dissimilarity distances to form clusters and the method set to complete linkages using the function hclust (“stats”, [30]), for all four pens. The clusters were then characterized according to the descriptive variables mentioned in Appendix A. To characterize the clusters according to the grouping of hens based on their movement patterns, generalized linear mixed effects models were applied to each cluster (level with 4 factors, based on the outcome of the cluster analyses) as a fixed effect and HenID (ranging between 103 and 108 hens per pen, respectively) as a random effect using the package lme4 [34], applying the parametric bootstrap approach in the pbkrtest package for model comparisons [35]. The residuals of the models were checked visually for normality and homogeneity of variance. The number of transitions was modelled using a negative binomial link function. Due to the presence of heteroscedasticity, robust linear mixed effects regressions (“robust LMM”, [36]) were used for the duration spent indoors. Using the rlmer function makes the estimation of the degrees of freedom non-trivial, and exact p values could not be computed directly [37]. The calculated t-statistics from the rlmer, as well as the estimated degrees of freedom of a regular lmer with the Satterthwaite method, were used to estimate an approximate level of significance from a default t-statistic table.

### 2.7. Timeshift Test for the Correlation between Social Association and Dynamic Time Warping

The clear negative correlation between dyadic association indices and DTW dissimilarity distances could be a result of a causal relationship between individual movement patterns and social affiliation, or it could be a confounding correlation as both the social associations and the movement patterns were derived from the same data: the detection of the birds at the antennas. In order to test whether the observed correlation was an artefact of data acquisition, we used a random sample of 14,400 dyads observed over an entire day and the associated time series of transitions of one of the two birds. For the selected bird, we shifted all entries by adding 60 s to the time stamps and re-calculated both association indices and DTW dissimilarity distance. In the case that both dyadic measures were intrinsically linked we would expect approximately the same correlations for the shifted data set as for the original data. Our findings confirmed this was not the case: a time shift of 60 s of one time series did not affect dyadic DTW dissimilarity distances (Pearson product–moment correlation between DTW scores based on original and shifted data r = 0.999), while it strongly affected the dyadic association (Pearson product–moment correlation between association indices based on original and shifted data r = −0.36), with the result that the correlation between dissimilarity distance and association indices for the sample of dyads rose from −0.085 to −0.057 (Appendix A).

## 3. Results

In total, we recorded 1,219,658 transitions of birds between two adjacent ranging areas. Due to missed detections, we could not locate the birds at all times, though on average (mean), a bird could be located 84.3% of the time. Use of outside areas varied substantially between birds. While ten birds (in pen 1 and 2: 2 birds each; in pen 3 and 4: 3 birds each, in total 2.4%) never left the barn and spent 100 percent of the observed time inside (we checked that tags were working and the birds were alive and well), others spent up to 87% of observed time (i.e., excluding portions of the day when the pop-hole was closed) outside of the barn (Figure 2 and Appendix A). We also observed differences in the use of outdoor areas, for example 33 birds (pen 1: 5.5%, pen 2: 12.6%, pen 3: 7.5% pen 4: 5.8%, in total 7.8%) never entered the free-range area. Thus, across all four pens, the wintergarden was used by 97.6% of the birds, the stone yard was used by 95.5% and the free-range area by 92.2%. On average (mean) birds spent 38% of the time in the wintergarden (pen 1: 33%, pen 2: 45%, pen 3: 40%, pen 4: 33%), 7.2% in the stone yard (pen 1: 7.7%, pen 2: 8.5%, pen 3: 6.8%, pen 4: 6.0%), 5.7% in the free-range area (pen 1: 8.5%, pen 2: 1.1%, pen 3: 4.5%, pen 4: 8.5%) and the rest of the time indoors.

We observed that, at the group level, transitions occurred often in small ‘bursts’ with groups of hens moving between two adjacent areas within a short time period (Figure 3a). Comparing the distributions of observed intervals between two hens passing from one area to the next with the expected distributions, assuming independence of the transition events, we found higher numbers of short intervals for all pens, indicative of co-ordinated movement between adjacent areas (Figure 3b and Appendix A). Kolmogorov–Smirnov tests suggested that the observed distributions of gap-times differed in all cases significantly from expected distributions (=un-coordinated random movement (Appendix A)).

### 3.1. Comparing Movement Patterns within and between Hens

For each hen and tracking day, on which all pop-holes and gates were open, the movement patterns could be extracted as daily time series of the individual transitions from one outdoor area to another (Figure 4a), resulting in 30,312 single-day time series that could be compared.

We used DTW to estimate the dissimilarity between all daily time series within each pen to compare each of the 72 daily time series for each hen to each other per pen (7776, 7416, 7488 and 7632 comparisons, for pen 1 to 4, respectively).

For each pen, the dissimilarity distances of the pairwise comparisons of the daily time series were smaller for within-hen comparisons (each hen compared with itself across time) than between-hen comparisons (each hen compared with all other hens across time, Figure 4b). This indicates a highly consistent movement pattern across time within hens and different movement patterns between hens (Wilcox Signed Rank Test: V = 0; *p* < 2.2 × 10^−16^; V = 0; *p* < 2.2 × 10^−16^; V = 1; *p* < 2.2 × 10^−16^; V = 6; *p* < 2.2 × 10^−16^ for pen 1, 2, 3 and 4, respectively).

To study the long-term trends in changes in daily DTW distances, we created generalized additive models using the 72 tracked days. The estimated change in long-term trend for the dissimilarity distances from day 1 to day 72 considering the entire tracking period of 166 days was very similar for each pen and slightly decreased by 1.58 % but was not significantly affected by day (F = 3.77; R^2^ = 0.038; *p* = 0.056; F = 3.17; R^2^ = 0.030; *p* = 0.080; F = 1.33; R^2^ = 0.005; *p* = 0.254; F = 1.56; R^2^ = 0.026; *p* = 0.158 for pen 1, 2, 3 and 4, respectively). Because there was no significant long-term trend at the hen level over the 72 tracking days, this indicates the stability of daily time series for individual hens over the entire study period.

### 3.2. Cluster Characteristics of the Movement Patterns Per Pen

To determine grouping of dissimilarity distance matrices, we summed up the daily dissimilarity distance matrices for individual hens in order to generate a matrix with one distance value for each within-hen comparison of the time series. Hierarchical clustering analyses using the Diana method available in the dtwclust function (divisive analysis, [31]) of these dissimilarity matrices suggested the existence of clusters of several distinct daily movement patterns in each pen. According to the clustering analysis on a pen level, we identified four main clusters in pens 1 and 3 and five clusters in pens 2 and 4 with the fifth cluster containing only one or two individuals, respectively (Appendix A). Two clusters showed congruent movement patterns across pens. In all pens, there was one group of hens clustering together with low transition rates and most time spent indoors and one group of hens clustering together with high number of transitions across outdoor areas and with a large fraction of their time spent in the free-range area (Figure 5a–c and Appendix A; Appendix A). The other clusters showed more pen-specific cluster characteristics.

### 3.3. Social Network Analyses

Social association matrices were calculated both for daily co-occurrence networks as well as for the overall co-occurrence recorded at the antennas for the entire observation period. The distributions of observed pairwise (dyadic) association strength values are highly skewed, with many dyads showing lower association values than expected for random co-occurrences, while a small number of dyads showed much higher strength values (Figure 6a and Appendix A), suggesting the existence of non-random social associations. The Newman–Girvan algorithm [38] for community detection identified between four and six communities per pen, though the assortativity by community was not very pronounced with assortativity indices ranging from 0.084 to 0.121 (Appendix A). At the individual level, hens showed considerable variation in PageRank centrality measures (Appendix A).

We investigated the temporal stability of the association networks by calculating auto-correlation coefficients for the daily association networks from day 1 to day 40 for time lags of 1 to 31 and by plotting the observed correlation coefficients as well as the expected correlation coefficients under the null assumption of random associations (Figure 6b). In all four pens the auto-correlation of association matrices did not decrease with increasing time lag, indicating highly stable social associations.

Correlating the overall matrix of dyadic association indices integrated over the entire observation period with the summed dyadic DTW dissimilarity distance matrix, we found strong negative correlations for all four pens. Matrix permutation tests [39] with 10.000 repetitions suggest that such high correlations values are unlikely to occur by chance (Appendix A; pen 1: r = −0.57, expected CI_95_ = −0.127–0.108, σ = 10.37; pen 2: r = −0.58, expected CI_95_ = −0.127–0.128, σ = 8.87; pen 3: r = −0.58, expected CI_95_ = −0.116–0.117, σ = 9.84; pen 4: r = −0.55, expected CI_95_ = −0.104–0.105, σ = 10.23). That is, hens that were more closely associated in the social network, were also more similar in their daily movement patterns as described by the DTW dissimilarity distance.

Matrix correlations of daily association matrices with daily DTW distance matrices were for all pens in almost all cases negative, indicating that more closely associated individuals were also more similar in their movement patterns (Figure 6c). Linear regressions suggest that the absolute value of the correlation coefficients increases over time in all four pens (pen 1: F = 9.7, df = 1, *p* = 0.027, R^2^_adj_ = 0.11; pen 2: F = 9.70, df = 1, *p* = 0.071, R^2^_adj_ = 0.032; pen 3: F = 14.55, df = 1, *p* = 0.0003, R^2^_adj_ = 0.16; pen 4: F = 29.14, df = 1, *p* = 8.7 × 10^−7^, R^2^_adj_ = 0.28, combined probabilities for all four pens: Χ^2^ = 56.65, df = 8, *p* = 2.1 × 10 ^−9^).

Comparing the cluster categorization of hens based on movement patterns and social associations, we found high correlations between the hens’ clustering based on DTW dissimilarity distances and the hens’ grouping based on social network communities for all four pens (Figure 7; likelihood ratio G-test: pen 1: Χ^2^ = 89.2, df = 12, *p* = 7.1 × 10^−14^; pen 2: Χ^2^ = 61.7, df = 9, *p* = 6.3 × 10^−10^; pen 3: Χ^2^ = 88.3, df = 12, *p* = 1.0 × 10^−13^; pen 4: Χ^2^ = 60.5, df = 15, *p* = 2.1 × 10^−7^, combined probabilities for all four pens: Χ^2^ = 56.65, df = 8, *p* = 1.5 × 10^−37^).

## 4. Discussion

We found individual movement patterns of space use in hens which were consistent over an extended time period of time (166 days) but differed markedly between hens. The high consistency of hens in their daily routines—manifested in high similarity of a hen’s daily time series of movements between areas—is in line with results from a recent study, where Rufener et al. [25] found differential movement and location patterns of tier use of hens within a multi-tiered aviary barn. Hierarchical cluster analysis allowed the identification of groups of hens with similar movement patterns by distinguishing a small number of movement types and the use of temporal analysis to suggest that over time hens became more similar in their movement patterns.

At a basic level, our results provide intriguing results about how individual animals use the various external areas provided. It is generally believed that flock use of external areas follows an inverse relationship with flock size and density, i.e., greater range use can be expected with smaller flocks (reviewed in ref. [40]). By extension, greater range use is associated with a variety of health and welfare measures including fewer birds with keel fractures [41] and less feather pecking [42]. Despite being a relatively small flock and with most hens in our study accessing the free-range area (with only 7.8% of hens not using the free-range area), the duration spent in the free-range area was considerably short. A likely explanation for this contrast to the expected results (i.e., greater or longer range usage) is that most studies focusing on individual usage of external areas have free-range areas only without transitional areas like the wintergarden and stone yard [43,44,45,46,47]. Although additional work is needed to determine what mechanisms are operating, our findings indicate that inclusion of a wintergarden is an effective management tool to increase use of external areas. Wintergardens, which are common throughout Switzerland, though uncommon elsewhere, offer many of the resources we believe are beneficial for the animals’ health and welfare including fresh air and sunlight, foraging and dustbathing opportunities, and increased floor space. Although these resources are largely available in free-range areas, birds would also be exposed to predators, direct sunlight, rain, snow, and other factors which are likely unattractive to the birds and disincentivizing the use of a free-range area. Hence, access to a wintergarden may provide many of the benefits of external areas without the free-range detriments. Although wintergardens are already widely accepted within certain segments of laying hen production, our results strongly suggest their consideration in addition to, or possibly even as an alternative, free-range area. While additional work is needed to compare preferences and benefits of these different outdoor areas, the basis for our recommendation lies in the apparent preference of hens for the wintergarden over the stone yard or the free-range area and the ability for hens to perform species-specific behaviour [48]. Wintergardens also offer a management benefit as animals can be given access to an outdoor area in a manner that eliminates exposure to migratory fowl and exposure to pathogens such as highly pathogenic avian influenza. In contrast, nations where wintergardens are uncommon, are required to keep their birds inside the barn during contagious disease outbreaks, causing a shift from free-range to barn-reared housing (e.g., Egg marketing Commission Regulation 589/2008). Taken together, our work presents the use and benefits of external areas in a very different perspective.

We conducted a social network analysis based on proximity data assuming that frequent co-occurrences of hens next to the same antenna reflects social association. The analysis showed that hens kept in groups of 355 individuals formed stable social associations with other hens, frequently associating with specific individuals. Surprisingly, we found clear correlations between the adjacency matrix of the social network and the dissimilarity distance matrix from the DTW analysis, indicating that more closely (socially) associated individuals were also more similar in their movement patterns. Moreover, birds showing similar movement patterns were also frequently found in the same social network cluster. These discoveries, found in all four replicated pens, require further investigation as they suggested a direct link between social structure and fine-scale individual movement behaviour.

As both the social network and the transitions between areas were derived from the same observational data, we investigated whether the observed relationship could be an artefact due to a confounding variable. By resampling from time-shifted data, we destroyed the network information (creating networks that were even negatively correlated with the original networks based on the non-shifted data), while the movement patterns remained intact (seen in correlation coefficients close to one for non-shifted and shifted time series). Doing so, we could demonstrate the relative independence between the network data and the dyadic time series dissimilarity distances—if both measures were intrinsically linked, the simulated time shift should have affected them equally. We thus assume that the correlation between social association and similarity of daily movement patterns must have a biological underpinning, which invites further in-depth investigations.

Furthermore, we note that we could only fit a sample of all hens of a pen with RFID tags. In order to investigate whether this limitation affects network descriptors based on associations of tagged individuals, we repeatedly re-calculated network measures after down-sampling. We found that global measures were rather stable and robust against down-sampling and that with increasing sample size individual centrality seems to tend towards an asymptotic value only slightly below the values reported for our sample (Appendix A). Together, we take these findings as indication that sampling only about a third of the animals resulted in a social network that is rather representative of the entire flock.

Looking at transitions of hens between areas we observed an uneven distribution of transitions over time, i.e., there were times when more animals moved between two adjacent areas than during other times. However, we did not observe ‘waves’ of individuals engulfing large proportions of the population, and hens were not moving between areas as one cohesive flock but were usually dispersed over all four available areas. The reason for this moderate level of coordination might be found in an existing but relatively weak individual tendency for following others.

We observed that individual differences in daily movement patterns were relatively consistent over time. Reale and Dingemanse [49] have suggested that the consistency in behaviour, given as the variation of this behaviour observed in a specific individual in relation to the variation of the behaviour within the population, is indicative of the degree of specialization. Specialization can contribute to the creation of social niches and, by that, to the reduction of competition between group members. The coexistence of different behavioural phenotypes has been the subject of various studies—most of them focusing on the context of social foraging [50,51]. Group composition in terms of behavioural phenotypes may influence the performance of a group which then feeds back on the individual fitness of group members [49,52,53] but also affects the overall optimal group size and density. The clear correlation between membership in a network community and time series type at the level of individual hens, is exactly what we would expect as a consequence of specialization.

The need to maintain group cohesion is often cited as a factor constraining the expression of individual behaviour and forcing individuals to coordinate their activity with the activities of other group members [18]. Early models of group formation usually assumed that individuals need to have information about the geometry of the whole group in order to optimize their own behaviour [21]. However, individual-based models of collective behaviour have demonstrated that cohesive flock level movement can also be achieved if individuals coordinate their movement with a small number of conspecifics [20,54,55,56]. Our observations suggest that social relationships influence the behaviour of individuals and that coordination between associated individuals might act as a force enhancing group cohesion. It could be worthwhile for a future study to ask whether the temporal coordination of daily movement patterns that we observed can function as a mechanism that ensures a certain degree of group cohesion while simultaneously allowing for flexible fine-tuning of cohesion strength and group size.

Reported overall differences in the proportion of space use between birds [22,23,24], Ref. [57] provided the starting point for our investigation, as differential space use might have practical implications for animal health and welfare as well as for animal management. Poultry farms in Europe and North America are increasingly adopting large, open systems with flocks of tens of thousands of individuals, where animals can freely access various distinct areas, whereas the ancestors of domesticated chickens typically formed groups of two to 20 individuals [58]. While chickens develop a pecking order in small groups, a theoretical study and several empirical studies with broiler chicken and domestic fowls report a change from a high aggressive dominance relationship towards low aggressive alternative strategies with increasing group sizes [59,60,61,62]. A study by D’Eath and colleagues has suggested that groups containing more than 70–80 animals will adopt an alternative strategy due to limitations in recognizing animals and the level of aggression needed to maintain a dominance position within such large groups [63]. In the current study hens formed social associations despite being housed in groups of 355 individuals. These divergent findings might have several reasons; e.g., in contrast to D’Eath and Keeling [63] the hens in our study were housed together continuously and assessments were based on observations in the home pen rather than a test situation. Additionally, we observed the formation of smaller clusters within the flock with divergent movement patterns, suggesting that hens within our study may have been reverting to a more ‘natural’ group size within the larger hen flocks, hence facilitating the formation of individual social associations. Understanding the underlying mechanism of individual space use might help to improve welfare on a flock level, in particular if differences in space use were linked to the health status of the animals. More generally, greater knowledge as to how animals within these complex systems use the available areas would be helpful in designing environments that are better suited to benefiting the welfare of the animal’s and their health.

## 5. Conclusions

Fundamental models of animal social organization [64,65,66,67] propose that resources underpin the spatial distribution of individuals, which in turn determines social structure. Yet, social relationships themselves can be considered to be valuable resources, leading to selection of mechanisms ensuring their maintenance [11]. As such, social relationships may be important drivers of variation in behaviour, affecting movement and activity patterns across spatial and temporal scales. In our study we found consistent individual differences in ranging patterns with a small set of distinct ‘ranging types’. All birds were from the same breeding line and were, hence, genetically very similar. Additionally, all birds were hatched and reared in the same breeding facility under standardized conditions. Thus, understanding where these substantial differences in space use come from will be a challenge for the future. At the same time the observed correlation between social associations and movement patterns calls for further investigations into the nature of this relationship.

## Figures and Tables

**Figure 1 animals-12-00555-f001:**
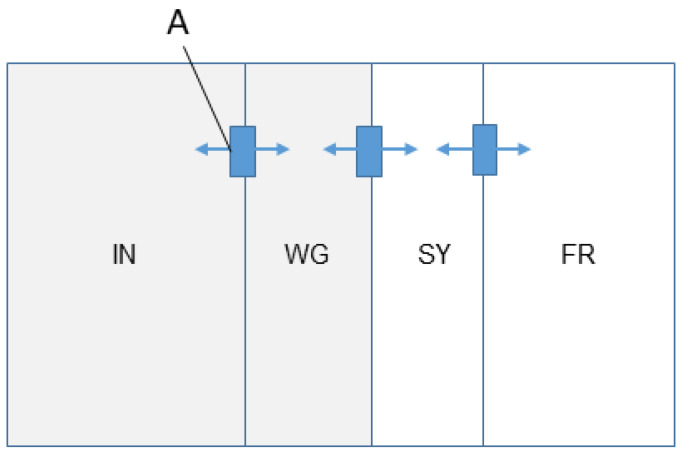
Schematic of a pen with four distinct areas. Each pen consists of an inside area (IN), a wintergarden (WG), a stone yard (SY) and a free-range (FR) area. Spaces linking areas were fitted with RFID antennas (A) that registered the presence of individuals and thus movements between areas. The inside area is fully enclosed and the wintergarden is covered by a roof (shaded area).

**Figure 2 animals-12-00555-f002:**
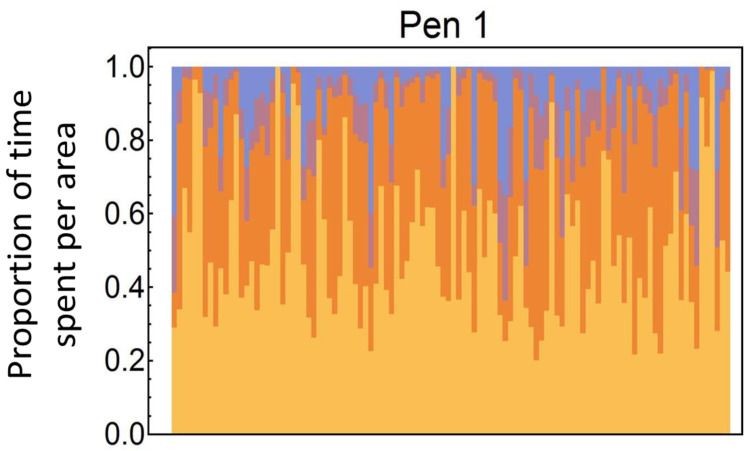
Stacked bar charts indicating individual space use of birds in one pen. Each column depicts the proportion of time a single bird spent in each of the four areas: indoors (yellow), wintergarden (orange), stone yard (violet), and free-range (blue). Charts for the other three pens are presented in Appendix A.

**Figure 3 animals-12-00555-f003:**
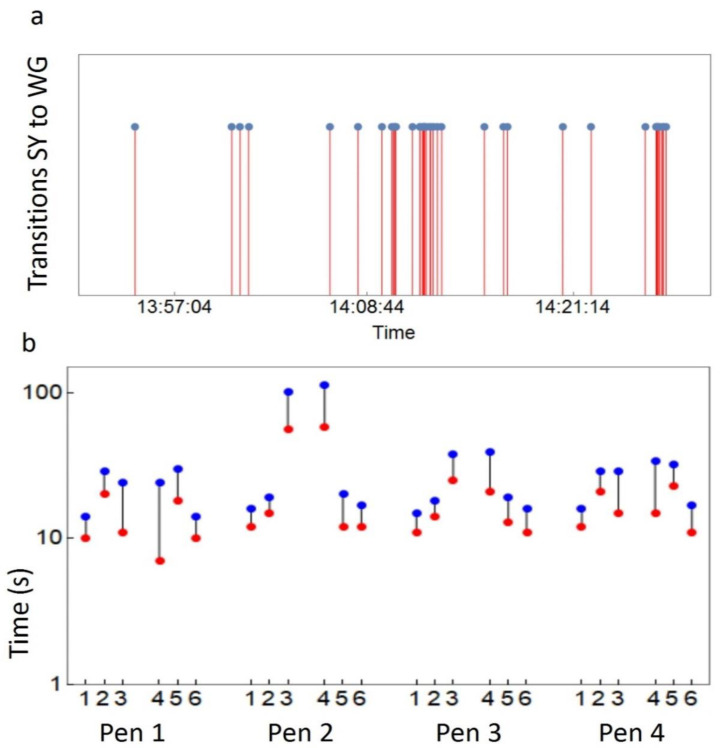
Transitions between areas. (**a**) Example time stream of transitions between SY and WG in pen 1 on 7 June 2016. Dots indicate transitions of individual birds at one specific gate. (**b**) Co-ordinated movements of hens between areas. Expected (blue) and observed (red) median gap times (s) between transitions of two birds from one area to another; 1: IN→WG, 2: WG→SY, 3: SY→FR, 4: FR→SY, 5: SY→WG, 6: WG→IN.

**Figure 4 animals-12-00555-f004:**
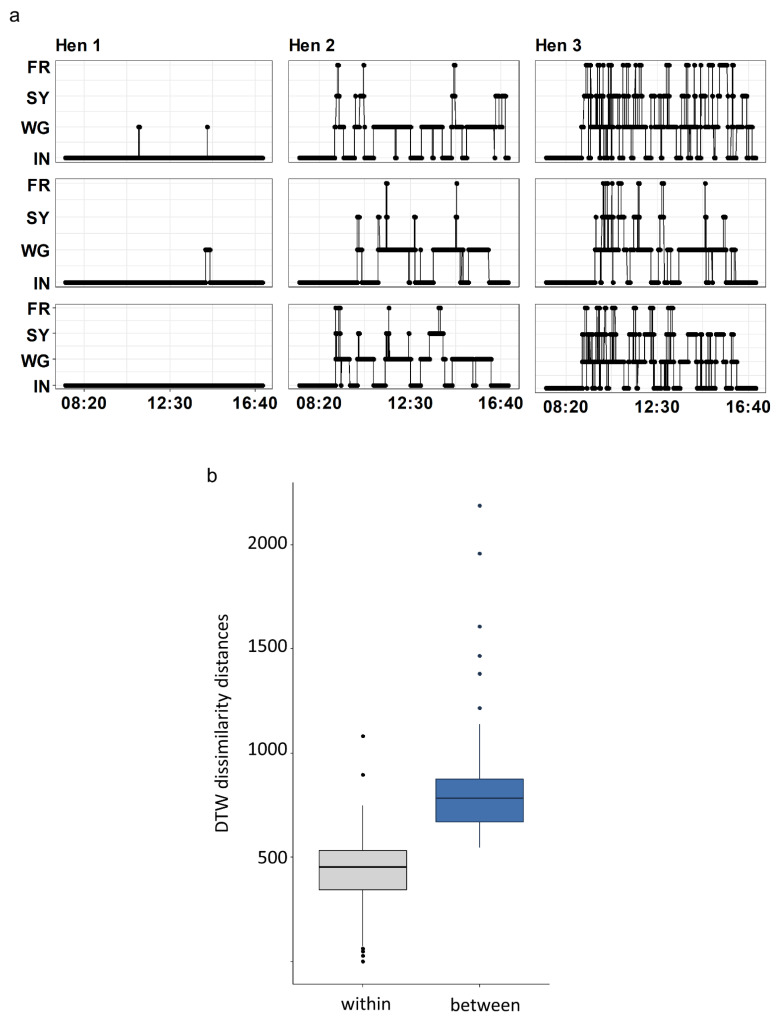
Daily time series. (**a**) Three daily time series (from top to bottom: day 1, day 36 and day 72) for three different hens that demonstrate the range of pattern differences that could occur within and between hens. The line indicates the presence of the hen in the given area. (**b**) Within (grey) and between (blue) hen comparison of daily time series DTW dissimilarities. Extreme values were not excluded because they reflect the valid within and between hen comparison of hens that never or hardly left the inside area.

**Figure 5 animals-12-00555-f005:**
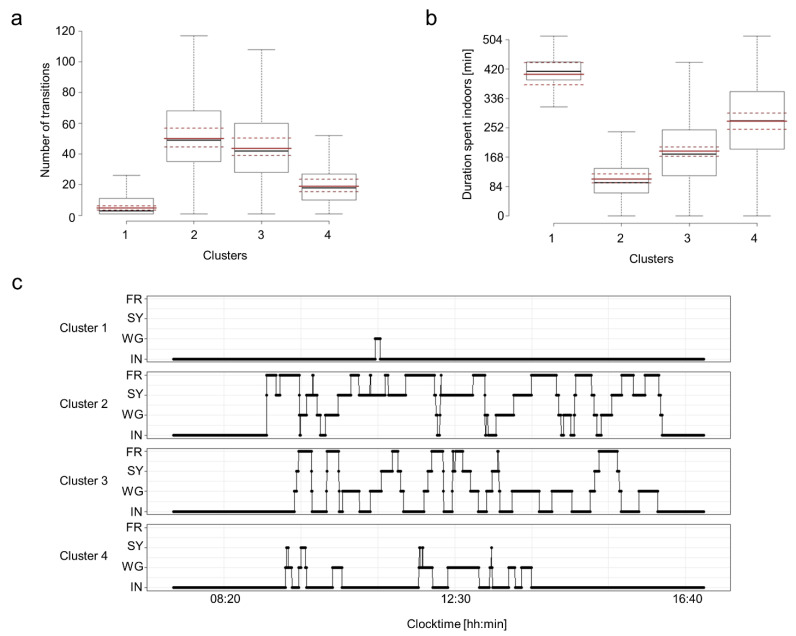
Characteristics of clusters for pen 1 based on the dissimilarity distances of the time series. Number of transitions (**a**) and duration spent indoors (**b**) per cluster. Solid red lines are model estimates, dashed red lines the 95-CI. (**c**) Typical daily time series patterns with the four zones (1: indoor; 2: wintergarden; 3: stone yard; 4: free-range) per cluster given for pen 1. Figures for pens 2–4 are given in the Appendix A.

**Figure 6 animals-12-00555-f006:**
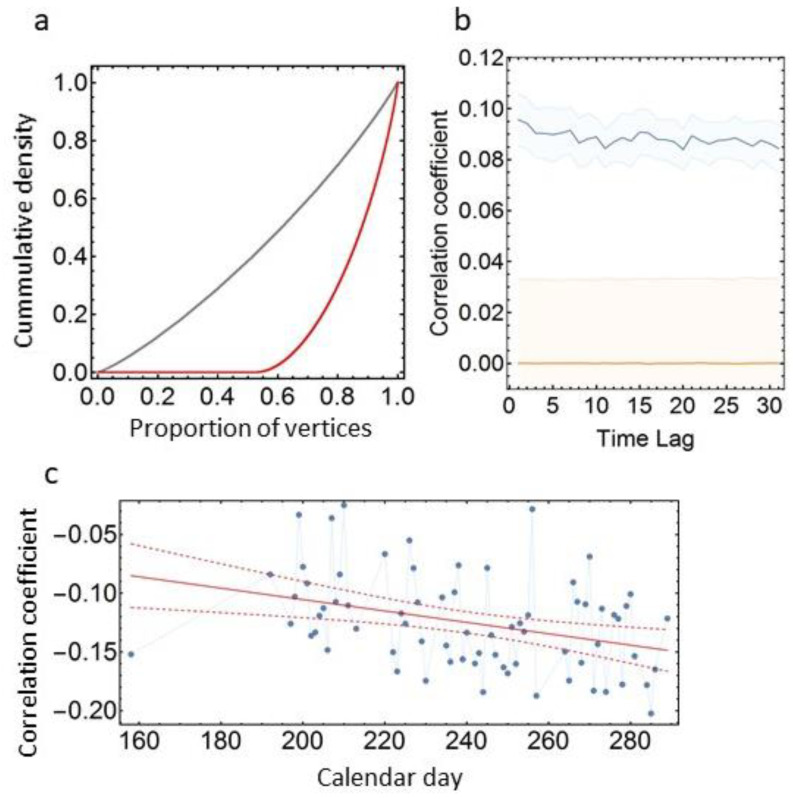
Dyadic associations. (**a**) Cumulative distribution function (CDF) for association strength of individual dyads summed over all observation days for pen 1. Grey: expected values based on random allocation of individual observations, red: observed values. CDF plots of association strength of the pens 2–4 are given in the Appendix A. (**b**) Averaged auto-correlation for daily association matrices for pen 1 for observational days 1 to 40 for time lags of 1 to 31. Blue: observed r with 95% CI (shaded area), orange: expected r for random associations based on random matrix permutation, orange shaded area: 95% CI for expected r. Auto-correlation plots of the pens 2–4 are given in the Appendix A. (**c**) Matrix correlation coefficients for daily association and DTW dissimilarity matrices for pen 1. Blue: observed correlation coefficients, red line: regression line of a linear regression model. Matrix correlation coefficients for pens 2–4 are given in the Appendix A.

**Figure 7 animals-12-00555-f007:**
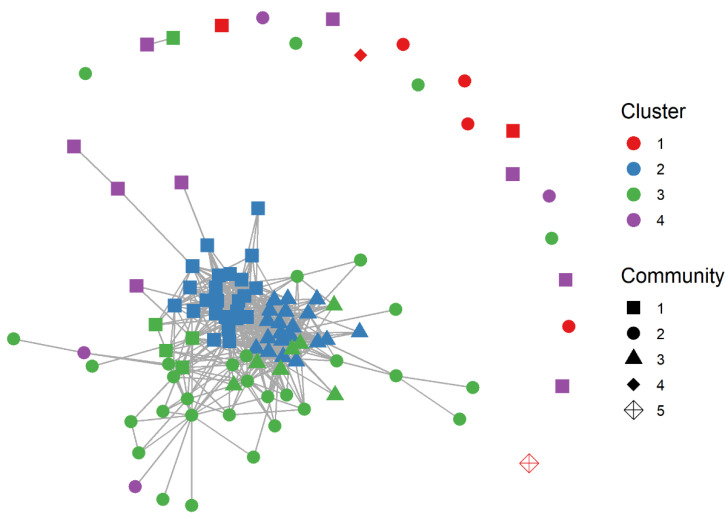
Social network graph of pen 1. Colours: clusters of movement and location patterns identified with DTW, symbols: network subgroups based on association index. For more details and for the other pens, see the link to the SHINYAPP (https://gomezya.shinyapps.io/shinyAppNetwork, last accessed: 13 February 2022).

## Data Availability

Data are available on https://gomezya.shinyapps.io/shinyAppNetwork (last accessed: 13 February 2022).

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
