# Peer review of "Similarity in Temporal Movement Patterns in Laying Hens Increases with Time and Social Association"

_animals, 2022, doi:10.3390/ani12050555_

Round 1
Reviewer 1 Report
The study aims to investigate whether hens formed non-random social associations that can be interpreted as individualized social relationships, also whether hens show consistent individual differences in their daily activity, and whether the similarities in daily activity patterns were correlated with social associations.
In general, the study is a novelty as it presents the social network analysis applied to laying hens. I made a few comments in the attached material. The most important is to change the `weather data` to `regional weather data` since the station was reasonably far away from the experiment. There is also a finding that needs some reference to support it (lines 472-473).
I found the supplementary material quite interesting, and it helps to understand the discussion.

Reviewer 2 Report
This is a really interesting paper that I have very much enjoyed reading. Use of space as, the authors indicate in the discussion, is closely related to social behaviour as actions in behaviour results in different movement patterns of individuals and groups. I would say that they feed on each other. In the literature, the number of papers in social behaviour is abundant, but on space use there are quite scarce. This is probably due to two reasons, the complexity involving data collection for space use, and even more so, the specific statistical analyses that are quite specific for space use. This work has resolved both in an elegant manner.
First, to have individually 1/3 of the population tagged with RFID tags (around 110 per pen x 4 pens) it is quite impressive. The authors in their discussion argued about the implications of not having every bird tag, but data collected on a third of the population should well represent the social dynamics of the group. Although the authors refer to an already published manuscript, it would be advisable include a brief description on how the ID tags were placed on the birds.
Regarding the statistical analyses, several of the calculated statistics were new to me and quite complex. Thus, I am afraid I am not able to judge the technical abilities of the authors in resolving such analyses. Nevertheless, the explanations that the authors gave regarding what they were trying to accomplish with such complex (apparently at least) statistics, together with my basic understanding make me feel quite comfortable with the approach taken. Also helps the fact that they also include other standard analyses such as proportion of times birds spent in the different study areas, the number of transitions across different areas etc, that also gave relevant information on the movement patterns and associations of the birds and are easier to understand calculations. I also liked the thorough consideration of the authors on considering potential problems of using the same data sources for the DTW and the dyadic association.
The manuscript is well written and structured, which make the reading of it a real pleasure. Perhaps a criticism that I can bring is the fact that some results are not discussed, especially in regard to the percentage of use of the different areas. It seems to me that the proportion of time spent in the free range areas was quite low. This has major implications from the practical stand point as it is often mentioned that use of free range is higher in small as compared to large groups, but these results do not show a high use of such area. Nothing regarding this result is mentioned in the discussion. In general, given the extension of the results of this nice study I felt that the discussion, although was very interesting and raised new aspects to consider, was cut short. I really liked the idea of the different behavioural phenotypes and the effects this may have on the individuals and the group at large.
A few specific points ;
L 32 ´We´should start a new sentence here
L 224 Were is this formula coming from? I imaging there should be a reference for this.
L233 Ecuclidean distances are the strait distances between two points. I cannot understand the relationship with DTW which seems to be a far more complex method to measure dissimilarities among two time series.
L317 I would like to see the mean proportion of birds and proportion of times the birds were using the different areas for the 4 pens within the paper. Perhaps adding a small table with the descriptors would be sufficient. The description of N birds that never enter the free-range area was, mean for one of the pens? Would be nice to have the % of birds not using the free-range for all pens.
L 328 Is there any statistical analyses that could be done to show coordinated movements? Are there statistical differences between expected and observed? In figure 3 b I am not really sure on how to interpret the results. Sorry….
L524 The idea of alternative strategies in the domestic fowl was suggested originally by Estevez et al., 1997 and in additional papers by Estevez et al in 2002 and 2003. Pagel and Dawkins published the mathematical model of alternative strategies in 1997.
The amount of data collected in this manuscript is massive and the results are quite impressive. Much of these results are included on supplementary information. However I feel that part of them should be on the manuscript, as it helps to understand concept and results. Specifically Fig. S4 and S5.
